# Dengue Fever Epidemics and the Prospect of Vaccines: A Systematic Review and Meta-Analysis Using Clinical Trials in Children

**DOI:** 10.3390/diseases12020032

**Published:** 2024-02-06

**Authors:** Ebele C. Okoye, Amal K. Mitra, Terica Lomax, Cedric Nunaley

**Affiliations:** Department of Epidemiology and Biostatistics, College of Health Sciences, Jackson State University, Jackson, MS 39217, USA; j00944302@students.jsums.edu (E.C.O.); j00412405@students.jsums.edu (T.L.); j00091335@students.jsums.edu (C.N.)

**Keywords:** dengue fever, clinical trials, case-cohort studies, children, vaccine, PRISMA, meta-analysis

## Abstract

About half of the world’s population is at risk of dengue infection. Epidemics of dengue fever have caused an increased risk of morbidity and mortality in recent years, which led to the exploration of vaccines as a preventive measure. This systematic review and meta-analysis aimed to evaluate the efficacy, immune response, and safety of dengue vaccines in children by analyzing clinical trials. The review followed standard procedures for data extraction using PRISMA guidelines and searching multiple databases, including PubMed, CINAHL, Medline, Health Source, Science Direct, and Academic Search Premiere. Eligible studies involved children (0–17 years old). Quality assessment was analyzed using the Cochrane Collaboration criteria, while data synthesis was conducted using thematic analysis and meta-analysis. Among the 38 selected studies, dengue vaccines showed varying efficacy against all four serotypes. The CYD-TDV (Dengvaxia^®^) and Tekade (TAK-003) vaccines showed strong protection against severe dengue, but their long-term efficacy varied. Vaccines triggered satisfactory immune responses, notably in those previously exposed to dengue. Safety profiles were mostly favorable, noting mild adverse events post-vaccination. Meta-analysis supported vaccine efficacy and immune response, but safety concerns warrant further exploration. In conclusion, dengue vaccines showed promising efficacy and immune response, particularly against severe manifestations.

## 1. Introduction

Dengue fever, caused by the dengue virus (DENV), is a significant global health concern due to its widespread prevalence and impact on public health [1]. It belongs to the Flavivirus genus and is primarily transmitted to humans through the bites of infected female Aedes mosquitoes, notably Aedes *aegypti* and Aedes *albopictus* [2]. The World Health Organization (WHO) characterizes dengue fever as an acute mosquito-borne viral infection characterized by fever, severe headache, joint and muscle pain, skin rash, and bleeding tendencies [3]. DENV exists in four distinct serotypes (DENV-1 to DENV-4) globally, while DENV-5 was first detected in the blood of a patient in the Sarawak state of Malaysia in 2007 [4]. Each stereotype can cause a range of clinical manifestations, from mild dengue fever to more severe forms like dengue hemorrhagic fever (DHF) and dengue shock syndrome (DSS) [5]. Severe cases of dengue can lead to complications like plasma leakage, potentially resulting in shock, coagulopathy, or vital organ impairment [6]. Globally, over the last fifty years, dengue fever infections have surged 30-fold to 390 million annually, with 96 million symptomatic cases and approximately 3.9 billion people across 129 countries at risk [7]. The disease prevails prominently in tropical and subtropical regions, where favorable climatic conditions sustain mosquito vectors, enabling year-round transmission [8]. The recent spread of dengue fever has been linked to various factors, including global trade and travel, population growth, and urbanization [9,10].

Recently, a notable dengue outbreak occurred in Bangladesh between January and August 2023, where a total of 69,483 dengue cases were reported, resulting in 327 related deaths (with a case fatality rate of 0.47%) [11]. Remarkably, the reported dengue cases for 2023 are the highest compared to the same periods recorded since 2000 [11]. Between 2010 and 2019, dengue outbreaks surged across North and South America. In 2010, over 1.6 million cases were reported, with 49,000 severe cases. The largest outbreak hit the US in 2016, with 2.38 million cases, mainly in Brazil. By 2019, the United States of America had a drastic rise, surpassing 3 million cases [12]. In the first half of 2023, South America had significant dengue outbreaks, with all four virus serotypes detected. Brazil, Colombia, Costa Rica, Guatemala, Honduras, Mexico, and Venezuela had all serotypes circulating, while other countries had varying combinations. The region reported nearly 3 million cases and 1302 deaths by 1 July 2023, with Brazil leading in cases (2.3 million), followed by Peru (188,326) and Bolivia (133,779) [13]. These outbreaks underscore the urgent need for effective preventive measures, especially vaccines, amidst multifaceted challenges.

Developing an affordable, safe, and effective dengue vaccine against the four DENV serotypes would be a significant advancement in controlling the disease, potentially aiding in achieving the WHO’s goal of reducing dengue morbidity by at least 25% and mortality by at least 50% [14]. Other studies have reviewed the spectrum of DENV vaccine development in detail, describing many different platforms, encompassing inactivated and live-attenuated virus vaccines, chimeric and viral-vectored vaccines, DNA vaccines, nucleic-acid-based vaccines, and subunit-based vaccines [15,16,17,18]. Dengvaxia (CYD-TDV) was the first available tetravalent DENV vaccine developed by Sanofi Pasteur and was initially licensed by several endemic countries [19]. It is aimed at ages 9–45 or 65 years old, providing a medical intervention against dengue. In addition, Ferguson et al. [20] developed a model that, considering routine vaccination at 80% coverage of individuals between 2 and 18 years old, there was an expected reduction of 20 to 30% in both symptomatic disease and hospitalization in high-transmission settings. In recent clinical trials, Dengvaxia only showed efficaciousness in individuals with pre-existing immunity to DENV. Seronegative individuals have an increased risk of severe disease after vaccination, making it contraindicated for naive populations [21,22,23]. Other important live-attenuated virus vaccine candidates currently being evaluated in clinical phase III trials are DENVax-TDV by Takeda and the TetraVax TV003/TV005 by the National Institutes of Health (NIH). TDV contains a chimeric-attenuated DENV2 strain that expresses the pre-membrane protein (prM-E) proteins of the other serotypes. TV003/TV005 is a combined formulation of four attenuated wild-type DENV serotypes. These vaccines have produced encouraging results in phase II trials, but long-term protection data are not available yet [24,25].

The significance of this study lies in understanding and addressing the global impact of dengue by analyzing outbreaks, evaluating vaccine prospects, guiding policy decisions, advancing research, and advocating comprehensive strategies to combat this widespread disease effectively. This systematic review and meta-analysis aimed to evaluate the efficacy, immune response, and safety of dengue vaccines in children by reviewing clinical trials.

## 2. Materials and Methods

### 2.1. Protocol

The Preferred Reporting Items for Systematic Reviews and Meta-Analyses (PRISMA) guideline [26] was used as a framework to guide the systematic review. This systematic review encompassed various stages: defining keywords, database searches for article selection, critical evaluation of studies, data selection and analysis, and presenting and interpreting results. Its research protocol is registered with PROSPERO, and the registration number is CRD42023478226.

### 2.2. Eligibility Criteria

The inclusion criteria include original, peer-reviewed journal articles, articles written in English, and studies that examined the immunogenicity, safety, and efficacy of the dengue vaccine in children aged 0 to 17 years. There were no limitations on the publication year and the country where the studies were conducted. Exclusion criteria include non-English articles, literature reviews, systematic reviews, book chapters, and conference papers. Also excluded were dengue studies that involved animal studies instead of human beings.

### 2.3. Search Strategy

Article searches were conducted across various databases, namely PubMed, CINAHL, Medline, Health Source, Science Direct, and Academic Search Premiere, from September to October 2023. Various combinations of keywords were used to search the electronic databases, including such terms as dengue outbreaks, vaccine development, vaccine efficacy, vaccine safety, human studies, children, etc. Additionally, further searches were manually conducted (including scanning reference lists) to identify articles that might not have been included in the initial search strategy. The search terms and Boolean operators utilized during the search strategy are shown in Table 1. 

### 2.4. Study Screening Process

The initial database search generated 1961 articles, and an additional manual search contributed 28 articles, which resulted in 1989 articles that were screened by the authors independently. From the initial pool of 1989 articles, 540 articles were found as duplicates and removed. Subsequently, the remaining 1449 articles underwent a two-phase screening process using an Excel sheet. In the first phase, titles and abstracts were screened, and 1388 articles were identified and removed. These articles were excluded because they did not meet the inclusion criteria and were irrelevant to this study, leaving 61 articles for further analysis. In the second phase, the remaining 61 articles were analyzed for full-text screening. Thirty-eight (38) articles were included in the systematic review, while 23 articles were excluded because the full text could not be accessed. For the subsequent meta-analysis, these 38 articles were further screened, resulting in the removal of 11 articles that lacked the data, thus not meeting the inclusion criteria. Therefore, a sum of 27 articles was included in the meta-analysis. The screening process is shown in the PRISMA flowchart shown in Figure 1 [26].

### 2.5. Quality Assessment

#### Risk of Bias and Quality of Evidence

The authors assessed the risk of bias in the clinical trial studies using the Cochrane Collaboration method for randomized trials (RoB2). The bias was judged on three levels: “high risk”, “low risk”, and “some concerns” (that is, when the provided information was not sufficient to make a clear judgment). The authors considered the following possible biases: blinding of participants and personnel (performance bias), random sequence generation and allocation concealment (selection bias), blinding of outcome assessment (detection bias), and selective reporting (reporting bias) [27]. 

The quality of the evidence was assessed using the GRADE approach. The studies were assessed for risk of bias, inconsistency between studies, indirectness, impression, and publication bias. According to the scoring criteria of the GRADE system, the rating for the quality of evidence was classified into four (4) levels: high certainty, moderate certainty, low certainty, and very low certainty.

### 2.6. Statistical Analysis 

The Review Manager^®^ 5.4 software was used to pool data using a random effects meta-analysis model. Subgroup analysis was done based on vaccine effectiveness and type. Analyses with heterogeneity, I^2^ > 40%, and a *p*-value for the X^2^ test < 0.10 [28] were considered to have substantial heterogeneity. The inconsistency (I^2^) method was used to assess heterogeneity among the studies. The effect size of the intervention was estimated by the total number of person-years using the risk ratio (RR) summary measure and the respective 95% confidence interval (95% CI). The efficacy, immunogenicity, and safety were estimated to be [1-RR] and were expressed as a percentage. A forest plot-type chart was used to present the results of the meta-analysis and the comparison of the studies. A funnel plot was plotted to determine publication bias. Egger’s regression test was used to assess the statistical significance of publication bias results.

### 2.7. Data Extraction

The authors developed a standardized form for data extraction with fields referring to the characteristics of identification of the studies, the countries involved, the study design, the publication year, and the participants’ ages (Table A1, Appendix A). Thematic analysis was used to identify, analyze, and present themes derived from the included studies [29]. The authors, who subsequently convened to reach a consensus on the themes chosen for result synthesis, independently identified the recurring themes from this analysis. 

## 3. Results

### 3.1. Study Subjects, Study Design, Locations, and Major Findings

A total of 1989 articles were screened for dengue vaccine efficacy, immunogenicity, and safety. Out of these studies, 38 articles met the inclusion criteria of this study. From the studies, the relevant characteristics were extracted, including information on the author, year of publication, study design, participants’ age, sample size, and countries where the studies were conducted (Table A1, Appendix A). Among the 38 studies analyzed, the majority, comprising 34 studies (89%), were conducted in Asian-Pacific and Latin American nations where dengue fever is prevalent. Four studies (11%) were conducted in South American countries (two in Brazil, one in Peru), and one in North America (the United States of America). In most studies, the age of the participants was stratified to mitigate bias. These studies were clinical trials [30,31,32,33,34,35,36,37,38,39,40,41,42,43,44,45,46,47,48,49,50,51,52,53,54,55,56,57,58,59,60,61,62,63,64,65,66,67], specifically centered on children aged between 0 and 17 years. The sample sizes varied significantly across the studies, ranging from 56 to 51,253 participants. The studies included 32 randomized controlled trials (RCTs) and six prospective case–cohort studies.

### 3.2. Outcome of the Risk Assessment

The risk of bias analysis using the Cochrane Collaboration tool [28] revealed that among the 32 randomized controlled trial (RCT) studies, 25 articles were rated as low risk in the following domains (information, confounding, selection, attrition, and reporting biases): [31,34,36,38,39,40,43,44,45,46,47,50,51,52,54,55,56,57,59,62,63,64,65,66,67]. Conversely, seven studies were rated as having “some concerns” about the risk of bias because of insufficient detailed information on the randomization process, such as blinding and concealment [30,32,33,53,58,60,61]. The risk of bias domains for each of the studies is displayed in Figure 2. 

### 3.3. Domains of Risk of Bias

The overall assessment of the risk of bias was low in over 75% of the studies (Figure 3). The following one area out of 5 domains raised some concerns in 7 out of 32 RCTs (22%): the specific area is “bias arising from the randomization process”. 

### 3.4. Vaccine Efficacy, Safety, and Immunonogenicity

The quality of the evidence of the outcome (vaccine efficacy, safety, and immunogenicity) for all the included articles was assessed using GRADEpro. Of all the included articles in the analysis, twelve articles were rated “high certainty” for efficacy because of the number of seropositive participants who were symptomatic after receiving the vaccines (CYD-TDV or Takeda). For the quality of evidence for immunogenicity, nine studies were rated “high certainty” because of the immune responses of antibodies by the seropositive participants. Six studies were rated “high certainty” for vaccine safety because of the seropositive participants that experienced adverse effects (AE) or severe adverse effects (SAE) (Table A2, Appendix B).

### 3.5. Effectiveness and Safety of Dengue Vaccines: Findings of Systematic Review

Among the 38 studies, 23 demonstrated the effectiveness of dengue vaccines in children involved in clinical trials. Most studies consistently showed the efficacy of both CYD-TDV (Dengvaxia^®^) and Tadeka (TAK-003) vaccines in preventing dengue fever caused by the four serotypes of the dengue virus [30,32,33,35,36,37,38,39,40,41,42,48,49,50,51]. However, a few studies reported discrepancies in their outcomes, revealing a lack of efficacy [31,34,52]. The effectiveness of these vaccines varied based on their formulations. While some vaccines displayed differing levels of efficacy against specific serotypes, others showcased broader protection encompassing multiple serotypes.

Out of the 38 studies examined, eleven confirmed children administered the dengue vaccine during clinical trials successfully triggered an immune response, particularly in the production of antibodies against the dengue virus [53,54,55,56,57,58,59,60,61,62,63]. Among these studies, seven focused on evaluating the immunogenicity of CYD-TDV in children, while the remaining four investigated the immunogenicity associated with the Tekade vaccine. Studies [54,56,57,59,60,62,63] illustrated a robust humoral response against all four DENV serotypes when CYD-TDV was administered to children through a three-dose regimen. Conversely, the four studies highlighted that the Takeda vaccine exhibited strong immunogenicity against all four dengue serotypes [53,55,58,61].

Out of the 38 studies reviewed, ten specifically addressed the safety profile of vaccines administered to children, particularly in regions where dengue is endemic [35,39,50,53,56,58,59,60,65,66,67]. These studies reported incidents such as hospitalization due to confirmed dengue cases, occasional deaths, and minor reactions such as rashes and headaches, mainly observed among both vaccinated and control children. These incidents were primarily documented in Asia and Latin America, categorized by the age of study enrollment and the study year.

### 3.6. Efficacy, Immunogenicity, and Safety of Dengue Vaccine: Findings of Meta-Analysis

A comprehensive meta-analysis was conducted, and data were pooled from 27 studies to evaluate the efficacy, immunogenicity, and safety of dengue vaccines. The total population generated from all the included studies was 145,452 and 71,869 participants in the intervention (vaccine) and control groups, respectively (Figure 4). The random effects model was utilized for the meta-analysis, and the result showed a heterogeneity (I^2^) of 94%, *p* < 0.00001, among the studies. This is considered substantial heterogeneity [28]. The analysis showed an overall risk ratio (RR) of 0.58 (42%), with a 95% confidence interval (CI) of 0.46–0.72.

A subgroup analysis was performed on the effectiveness of dengue vaccines (Figure 5). A random effects model was employed in the meta-analysis and the pooled effects of all the subgroups. Regarding the subgroup analysis of vaccine efficacy, 12 studies (out of the 27 included studies) contained data related to vaccine efficacy, including 89,498 and 45,242 participants in the intervention (vaccine) and control groups, respectively. The vaccine efficacy had a risk ratio of 0.62 (38%) with a 95% CI of 0.48 to 0.81 (*p* < 0.00001). However, the analysis showed an I^2^ of 95%, which shows considerable heterogeneity among the studies because the I^2^ > 40. Simultaneously, nine studies were evaluated for the subgroup analysis of immunogenicity. The total number of participants in the included studies was 30,499 for the intervention group and 13,896 for the control group. The RR for the immunogenicity was 0.48 (52%), with a CI of 0.25–0.91 and *p* < 0.00001, and a substantial–high I^2^ of 95%. For the subgroup analysis for vaccine safety, six studies were assessed, including 25,083 and 12,547 participants for the intervention and control groups, respectively. The meta-analysis showed that the RR was 0.75 (25%), with a CI of 0.46–1.22 and *p* = 0.005, with a high I^2^ of 70% across the studies.

In the subgroup analysis based on vaccine type (as depicted in Figure 6), a random-effect model was used for the meta-analysis, providing insights into the pooled effects for the respective subgroups. Two distinct vaccine types, CYT-TDV and TAK-003, were analyzed. For CYT-TDV, eighteen studies with a cumulative sample size of 114,947 were included in the analysis. The resulting risk ratio (RR) was 0.69 (31%). The 95% CI ranged from 0.56 to 0.84 (*p* < 0.00001), and the I^2^ within this subgroup was 89%. With TAK-003, nine studies with a total sample size of 102,374 were included in the meta-analysis. The calculated RR for this subgroup was 0.42 (58%). The 95% CI ranged from 0.27 to 0.67 (*p* < 0.00001). The heterogeneity within the TAK-003 subgroup was notably high, at 95%.

### 3.7. Funnel Plot: Publication Bias Results

The publication bias was assessed using a funnel plot to visually examine the distribution of the estimate of the effect size (Figure 7). Sixteen studies were symmetrically scattered on both sides of the risk ratio within the funnel. However, eleven (11) studies were scattered outside the funnel plot, which may indicate asymmetry and potential publication bias.

### 3.8. Egger’s Regression Analysis Results for Publication Bias

Egger’s regression analysis was conducted to assess the statistical significance of the results of publication bias and identify an asymmetry in the data (Table 2). The standard error and intercept values were 1.227 and −0.944, respectively. The two-tailed *p*-value associated with the intercept was 0.449, and the 95% confidence interval ranged from −3.470 to 1.582.

## 4. Discussion

In this study, the authors assessed the efficacy, immunogenicity, and safety of dengue vaccine candidates (CYD-TDV and Takeda) in children.

We found that almost all the included studies were rated as having a low risk of bias in the domains and the overall pooled effect. This showed that the overall pooled effect showed a reduced risk of severe dengue in the intervention group compared to the control group in terms of the outcomes (efficacy, safety, and immunogenicity). The observed reduction in the risk of severe dengue within the intervention group, as shown by the overall pooled effect, underscores the positive impact of the intervention in terms of efficacy, safety, and immunogenicity. This collective improvement across multiple outcomes suggests that the intervention not only effectively mitigates the occurrence of severe dengue but also shows a favorable safety profile and induces a robust immune response.

The result showed that the quality of the evidence conducted had an overall rating of “high certainty”. We hold strong confidence that the true effect lies close to that of the estimate of the effect. We recommend the administration of dengue vaccines (CYD-TDV and Takeda) to children aged 0–17 as a robust strategy for controlling dengue fever. This measure will aid in the prevention of dengue infection in children and also significantly contribute to the reduction of morbidity and mortality associated with dengue diseases.

We conducted forest plot analysis for all the included studies, using a random-effects model for the meta-analysis. Our results revealed a significant level of heterogeneity (I^2^ = 94%, *p* < 0.00001) among the included studies. This high heterogeneity surpassed the commonly accepted threshold of 40%, which showed substantial variability in study outcomes that could be because of the different age ranges of the participants or the vaccine formulations. For the subgroup analysis performed for the efficacy of the vaccines, we found that the risk ratio was less than 1 (0.62), which signified that the vaccines were efficacious in reducing the risk of severe dengue within the intervention group when compared to the control group. Also, the vaccine immunogenicity had an RR of 0.48, thus showing that the intervention/vaccinated group had a better immune response up to three doses compared to the control group. For vaccine safety, the RR was less than 1 (0.75), so the risk of adverse effects/events was 25% lower in the intervention group compared to the control group. This showed a potential protective effect or a lower likelihood of the event occurring. There was a significant difference observed among the six studies because the pooled effect crossed the line of the null effect (Figure 5). In comparison, the outcomes of all the subgroups showed that the intervention group was favored in terms of the vaccines’ effectiveness (efficacy, immunogenicity, and safety) over the control group. Also, all the subgroups had high heterogeneity of 95%, 95%, and 70% for efficacy, immunogenicity, and safety, respectively.

According to the subgroup meta-analysis by vaccine type, we found that both the CYT-TDV and TAK-003 subgroups exhibited a significant reduction in the risk of severe dengue in the intervention group, although risk reduction was more pronounced in the TAK-003 subgroup. We observed heterogeneity in both subgroups, which could be because of differences in vaccine formulations and dosage. This underscored the need for further investigation into the sources of variability among the included studies. The comprehensive analysis of different vaccine types will contribute valuable insights for understanding the nuanced effectiveness of dengue vaccines across various formulations. From the funnel plot analysis for publication bias, we visually observed and found that while most studies exhibit a symmetrical distribution within the funnel plot, asymmetry in eleven studies raises concerns about potential publication bias. Studies outside the funnel may imply selective reporting, where studies with specific characteristics, often those with statistically significant or positive results, are more likely to be published, while studies with non-significant or negative results may be underrepresented. We conducted Egger’s regression tests for publication bias to provide further statistical insights into the significance of the observed asymmetry. We found that the two-tailed *p*-value (*p* = 0.449) was not statistically significant, so there was no statistically significant evidence of publication bias in the analyzed studies. This showed that the observed asymmetry in the funnel plot was not statistically significant and could be attributed to random variation. This further strengthens the credibility of the meta-analysis findings, suggesting that the observed effects are not systematically influenced by selective reporting or publication-related biases.

Based on the results, we found that both vaccines are protective for children. Children who were previously exposed to dengue fever (seropositive) before vaccination exhibited superior outcomes in terms of immunogenicity, efficacy, and safety compared to those without prior exposure (seronegative). The clinical studies referenced in this review were registered with ClinicalTrials.gov.

In terms of efficacy, we found that dengue vaccines showed strong protection against severe virologically confirmed dengue (VCD) and hospitalization in children, but their effectiveness varied over extended follow-ups in clinical trials. TAK-003 proved effective against symptomatic dengue for three years, with sustained protection against severe cases despite declining overall efficacy. An ongoing study across eight dengue-endemic countries supported the TAK-003’s usefulness in controlling dengue [45]. CYD-TDV (Dengvaxia^®^) offered protection to children with prior dengue exposure for up to six years but posed higher risks to individuals without previous exposure during outbreaks [48]. While CYD-TDV (Dengvaxia^®^) and Tadeka (TAK-003) exhibited high efficacy in clinical trials against the four DENV serotypes, certain studies revealed efficacy biases because of the lower detectability of primary infections among vaccinated seronegative individuals [30,31,33,34,35,36,37,38,40,41,45,46,47,48,49,50,52]. There were variations and declines in efficacy, regardless of serotype or previous exposure, necessitating ongoing assessments of long-term vaccine performance [32,39,42,43,44,51]. According to Forrat et al [30]., CYD-TDV protected against hospitalization and severe VCD in participants aged ≥ 9 years and baseline seropositive over 6 years compared to placebo. Also, vaccine protection was observed over the different study periods, but the highest protection occurred during the first 2 years. During this period, a notable reduction in the risk of hospitalization and severe VCD was evident, particularly among seropositive participants aged 6–8 years. The overall efficacy increased with age among seropositive children [33], indicating that age is a factor when it comes to vaccine efficacy. Clinical signs and symptoms and quantified dengue viremia from participants with hospitalized VCD were comparable between groups [30]. Yang et al. also reported that CYD-TDV was efficacious for the participants. However, the vaccine showed higher efficacies for all serotypes (DENV 1–4) in baseline seropositive participants than in baseline seronegative participants, where it showed moderate efficacy only against DENV-4. This means that the serostatus of the participants should be taken into consideration when administering the dengue vaccine. Biswal et al. reported that TAK-003 was well tolerated and efficacious against symptomatic dengue in children, irrespective of serostatus before immunization, and that vaccine efficacy varied by serotype, hence suggesting the need for continued follow-up to assess longer-term vaccine performance [44,46].

In terms of immunogenicity, the meta-analysis showed that seropositive individuals who received CYD-TDV or Takeda showed better immune responses compared to seronegative individuals. These vaccines were immunogenic against all four dengue serotypes, irrespective of the baseline dengue serostatus. CYD-TDV showed neutralizing antibody responses against all dengue serotypes, with differences by age and endemicity that persist above baseline levels in endemic countries [57]. A study reported that a clinical trial showed the persistence of neutralizing antibody titers against TAK-003 over 3 years in children living in dengue-endemic countries, with a limited contribution from natural infection [55].

One key finding in examining dengue vaccine immunogenicity is the varying responses elicited by different serotypes of the virus. Dengue virus exists in four serotypes (DEN-1 to DEN-4), and vaccines aim to protect against all strains. However, studies have revealed that eliciting a balanced immune response against all serotypes remains a challenge. Some vaccines showed strong immunogenicity against specific serotypes, potentially leading to uneven protection and a risk of severe disease upon subsequent exposure to different serotypes. Additionally, the age of individuals who received the vaccines influenced their immunogenicity. Studies showed that younger age groups exhibited more robust immune responses compared to older individuals. This disparity in response among age cohorts has significant implications for vaccination strategies, causing the need for tailored approaches to ensure adequate protection across diverse age ranges. Research has explored the role of pre-existing immunity, particularly in areas where dengue is endemic. Individuals with prior exposure to dengue or those living in regions with high dengue prevalence may exhibit different immune responses to vaccination compared to those in non-endemic areas. Therefore, understanding how pre-existing immunity impacts vaccine-induced immune responses is crucial for optimizing vaccination strategies and predicting vaccine efficacy in different populations. Tricou et al. [58], who conducted a long-term clinical trial of TAK-003, reported that the vaccine elicited antibody responses against all four serotypes, which persisted to 48 months post-vaccination, regardless of baseline serostatus. Watanaveeradej et al. [60] reported that the results of two follow-up studies they conducted using the CYD-TDV showed that the live-attenuated DENV candidate vaccine did not elicit a durable primary humoral immune response. These variations in the duration of protective immunity among vaccine candidates (CYD-TDV and TAK-003) showed the need for booster doses or alternative vaccination schedules to maintain sustained protection against dengue.

Regarding vaccine safety, studies showed that the CYD-TDV and Tekade did not have any significant safety concerns, particularly among children with prior dengue exposure [35,39,50,54,56,57,58,65]. We found that seropositive individuals (children previously exposed to dengue) who received these vaccines exhibited better tolerance compared to seronegative children. This discrepancy is attributed to the former group experiencing fewer or milder adverse effects, whereas the latter were more susceptible to severe adverse reactions. Participants in the intervention group experienced common side effects, including mild-to-moderate fever, rash, headache, and myalgia, occurring within 12 days after the first (1st) dose and typically lasting for three (3) days or less [66]. Arredondo-García et al. [65] reported the overall relative risk in children aged < 9 years for Year 1 to Year 4 follow-up, with a higher protective effect in the 6–8-year-olds than in the 2–5-year-olds. According to Reynales et al. [48], the participants in the CYD-TDV group experienced at least one (1) serious adverse event (SAE), while 16.2% of the control group experienced SAEs, mostly related to infectious diseases. Out of the 29 deaths reported, 20 occurred in the CYD-TDV group and 9 in the control group. However, none of the deaths or SAEs were related to CYD-TDV (respiratory, thoracic, and mediastinal disorders and asthma). The deaths were attributed to traffic accidents, violence (i.e., gunshot wounds, stabbing, homicide), intentional self-poisoning, and exposure to other unspecified chemicals and noxious substances, etc. Hadinegoro et al. [39] conducted long-term surveillance spanning 3 to 6 years to monitor the safety of children who received the CYD-TDV dengue vaccine. They recorded a significant increase in hospitalizations, particularly observing an unexplained surge in dengue-related hospital admissions among children under 9 years old during the third year. This trend requires careful monitoring in long-term follow-ups. However, among children aged 2 to 16 years, the risk was lower in the vaccine group compared to the control group. There was a reduced risk of hospitalization for dengue overall for up to 2 years after completing the three-dose vaccination schedule among children aged 9 to 16 years. According to Sáez-Llorens et al. [53], vaccine-related unsolicited adverse events occurred in 14 (2%) out of 562 participants. However, no vaccine-related serious adverse events were identified. Rivera et al. [45] documented seven deaths during the clinical trials of Takeda’s vaccine, TAK-003, of which five occurred in TAK-003 recipients and two in placebo recipients. They also reported serious adverse events (SAEs) in 2.9% of those who received TAK-003 and 3.5% of those who received the placebo in the initial phase of Part 3. However, none of these deaths or SAEs were related to the study vaccine. Overall, no significant safety risks were identified throughout the study period. Biswal et al. [61] recorded no deaths or adverse effects (AEs), which did not lead to the withdrawal of the participants in the study. During the study, three participants reported four serious adverse events (SAEs): two occurred in the placebo group (both were moderate, then appendicitis and ankle fracture occurred after the second vaccination) and two in the TAK-003 group (both were severe, then abdominal pain and urinary tract issues occurred after the first vaccination). None of these SAEs were attributed to the trial vaccination or procedures, and none resulted in withdrawal from the trial or discontinuation of the vaccination process. Also, the dengue vaccine was well tolerated by the participants, with no serious adverse events or alert laboratory values. Although these SAEs reported in the intervention group were negligible, this might cause vaccine hesitancy among the parents of the participants (children). Studies have shown that most parents perceive vaccine side effects as more extensive than the information conveyed by their physicians, leading them to weigh the risks as outweighing the benefits of vaccinating their children [68,69].

### 4.1. Strength of the Study

The strength of the study lies in its comprehensive analysis of multiple clinical trials investigating dengue vaccines in children. It involves a systematic review and meta-analysis, incorporating diverse studies and allowing for a robust analysis of vaccine efficacy, safety, and immunogenicity. The study’s rigorous method, including adherence to PRISMA guidelines, rigorous risk of bias assessment, and GRADE approach for quality evaluation, enhances the reliability and credibility of the findings. Including many studies and participants contributes to the study’s statistical power.

### 4.2. Study Limitations

The study has limitations regarding its focus on English-language articles, potentially excluding valuable non-English research. The study concentrated on children aged 0–17 years, which may limit its generalizability to all populations. Most of the studies used for this study were conducted in dengue-endemic regions, which might limit the applicability of the findings to other geographic areas. Variability in study durations could affect the results’ consistency, while limited follow-up periods may impact long-term efficacy and safety assessments. Diverse vaccine formulations and dosages make direct comparisons challenging. Several ongoing but incomplete studies on some other dengue vaccines (e.g., TAK 005) could not be evaluated in this meta-analysis because of a lack of sufficient data. Another potential weakness of the studies reviewed here is that, although several studies reported the efficacy of the vaccines on dengue severity, including hospitalization, they did not report any biochemical responses such as cytokine profiles (an indicator of disease severity) following vaccination.

## 5. Conclusions

This systematic review and meta-analysis provide an extensive evaluation of dengue vaccines in children, explaining their efficacy, immune response, and safety. The study’s significance lies in its analysis of multiple vaccines, offering a comprehensive understanding of their performance in combating dengue fever. Notably, the study emphasizes the critical influence of prior dengue exposure on vaccine outcomes. Seropositive individuals exhibited more favorable responses to vaccination, with higher efficacy and improved safety profiles compared to their seronegative counterparts. These findings stress the necessity of personalized vaccination strategies, tailoring approaches based on individuals’ serostatus to optimize vaccine effectiveness and safety. The analysis showed varying efficacy levels among different vaccines. While some vaccines showed strong protection against severe dengue and hospitalization, efficacy fluctuated over extended follow-up periods in clinical trials. Continued evaluation of long-term vaccine performance remains imperative to understand their sustained efficacy, immune response, and safety. The study contributes novel insights into dengue vaccine development and deployment, guiding policymakers, healthcare professionals, and researchers in shaping targeted public health interventions. The findings call for ongoing vigilance in monitoring vaccine performance, especially regarding prolonged efficacy, safety, and population-specific responses. In conclusion, this study underscores the importance of nuanced approaches to dengue vaccination, considering serostatus, vaccine efficacy, and safety profiles. It serves as a pivotal resource for informed decision-making, advancing research, and advocating comprehensive strategies to combat dengue fever effectively.

## Figures and Tables

**Figure 1 diseases-12-00032-f001:**
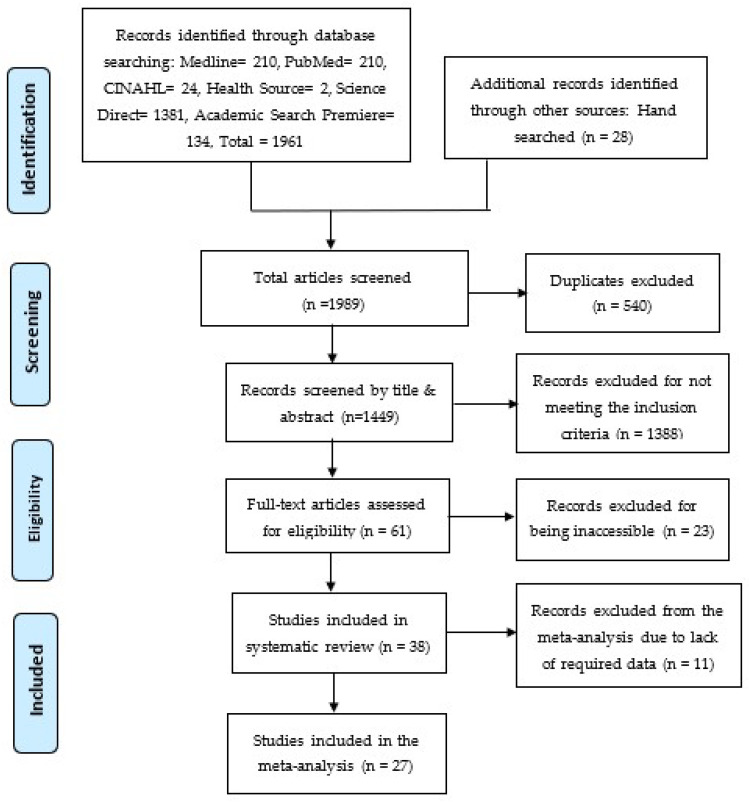
PRISMA Flowchart [26].

**Figure 2 diseases-12-00032-f002:**
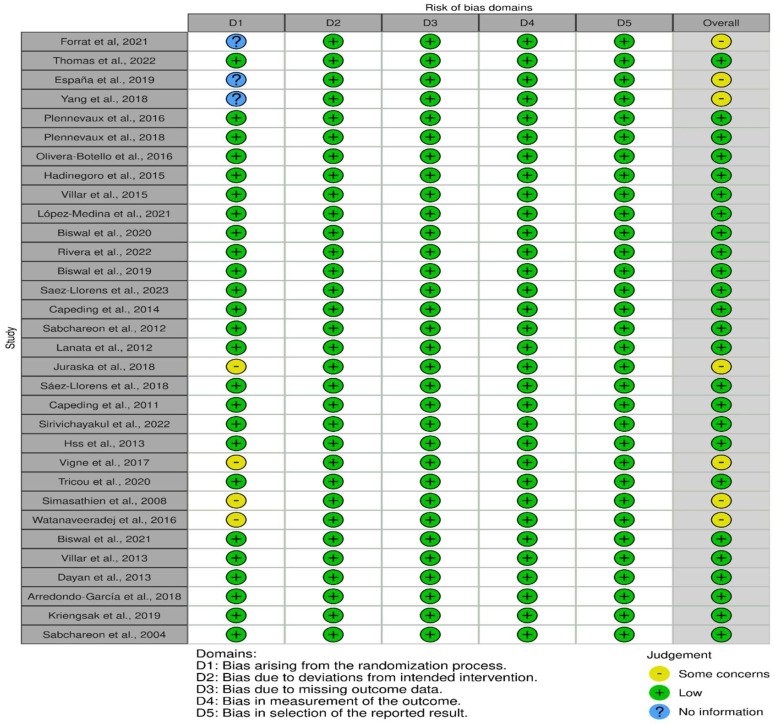
Risk of bias was presented in 5 domains, representing bias arising from randomization, bias due to deviations from intended intervention, bias due to missing data, measurement bias, and bias in selection of the reported results. The overall judgment presented as some concerns (yellow), low risk (green), and no information, as demonstrated in 32 studies [30,31,32,33,34,36,38,39,40,43,44,45,46,47,50,51,52,53,54,55,56,57,58,59,60,61,62,63,64,65,66,67].

**Figure 3 diseases-12-00032-f003:**
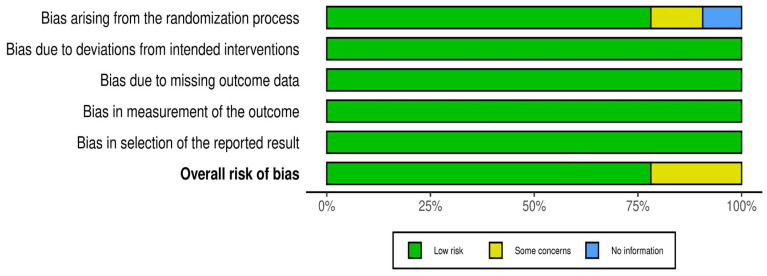
Overall distribution of risk of bias in 32 RCTs.

**Figure 4 diseases-12-00032-f004:**
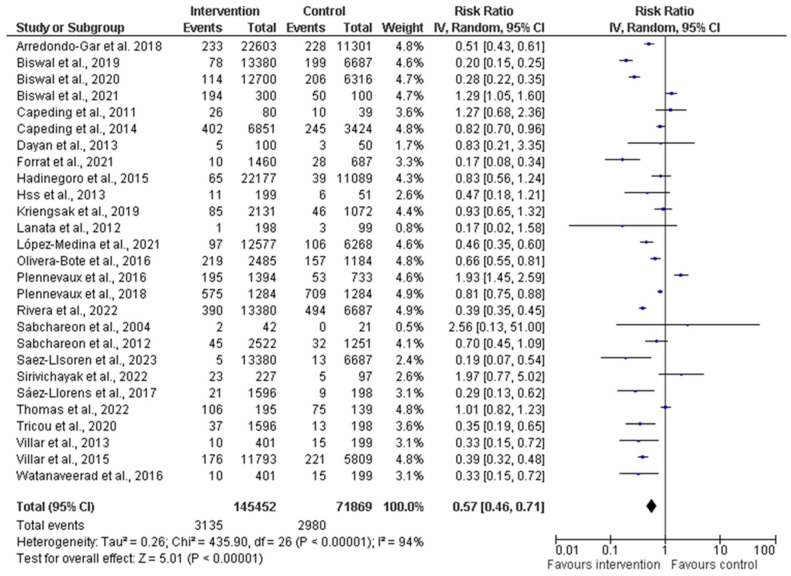
Forest plot showing pooled analysis of 27 studies by random effect model [30,31,34,36,38,39,40,43,44,45,46,47,50,51,53,54,55,56,58,60,61,62,63,64,65,66,67]. The vertical line indicates null or no effect. The horizontal lines indicate 95% confidence interval of relative risk. The horizontal lines that cross the null line show the study results are not statistically significant. The diamond shape ⧫ represents the overall effect of the summary of all studies. In this figure, it indicates favorable response toward intervention with the vaccines.

**Figure 5 diseases-12-00032-f005:**
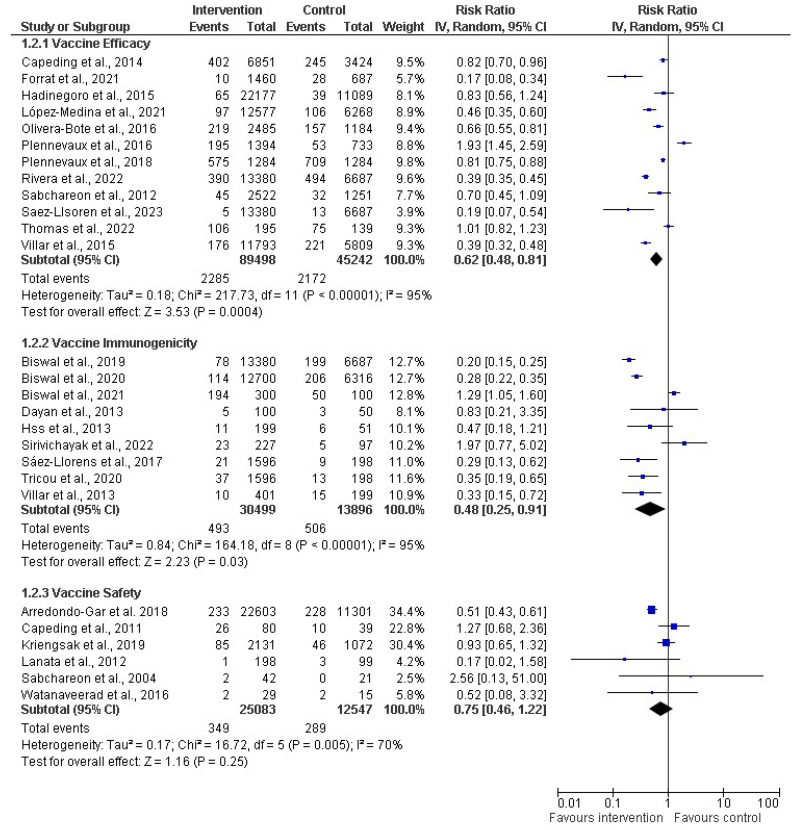
Subgroup analysis of dengue vaccines in children by vaccine efficacy (12 studies) [30,31,34,36,38,39,40,43,45,47,50,51], immunogenicity (9 studies) [44,46,53,55,56,58,61,62,63], and safety 6 studies) [54,60,64,65,66,67]. The vertical line indicates null or no effect. The horizontal lines indicate 95% confidence interval of relative risk. The horizontal lines that cross the null line show the study results are not statistically significant. The diamond shape ⧫ represents the overall effect of the summary of all studies. In this figure, it indicates favorable response toward intervention with the vaccines.

**Figure 6 diseases-12-00032-f006:**
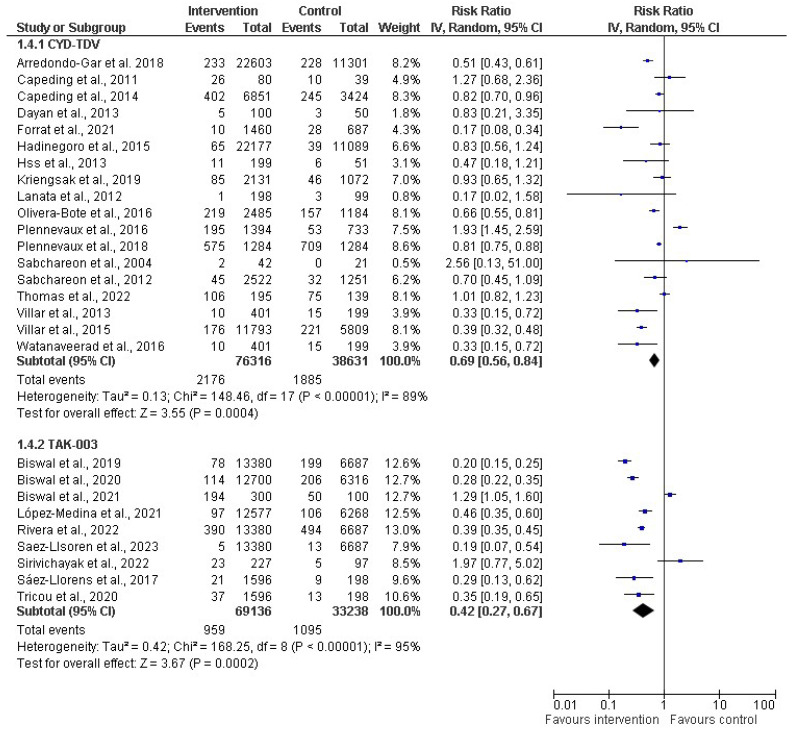
Subgroup analysis of dengue vaccines by type (CYD-TDV (Dengvaxia^®^) [30,31,34,36,38,39,40,50,51,54,56,60,62,63,64,65,66,67]; and Tadeka (TAK-003) [43,44,45,46,47,53,55,58,61]. The vertical line indicates null or no effect. The horizontal lines indicate 95% confidence interval of relative risk. The horizontal lines that cross the null line show the study results are not statistically significant. The diamond shape ⧫ represents the overall effect of the summary of all studies. In this figure, the overall effect indicates favorable response toward intervention with the vaccines.

**Figure 7 diseases-12-00032-f007:**
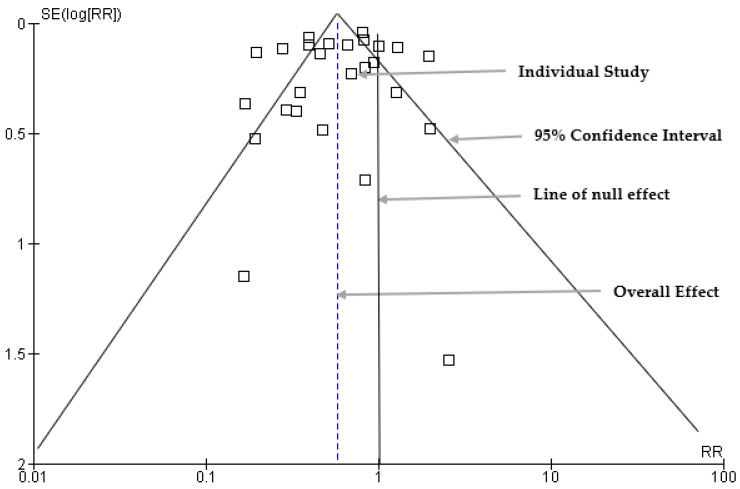
Funnel plots for assessment of publication bias.

**Table 1 diseases-12-00032-t001:** Databases searched, the keywords utilized, and the number of articles.

Databases	Search Keywords	Number of Articles Found
PubMed (1)	“Dengue fever” OR “Dengue epidemics” OR “Dengue vaccine” OR “Dengue Vaccine prospects”	129
PubMed (2)	“Dengue fever” OR “Dengue epidemics” AND “Dengue vaccine” OR “Dengue vaccine development” OR “Dengue vaccine prospects” AND “Dengue vaccine efficacy” OR “Dengue vaccine safety” OR “Dengue serotypes”	81
CINAHL	“Dengue fever” OR “Dengue epidemics” AND “Dengue vaccine” OR “Dengue vaccine development” OR “Dengue vaccine prospects” AND “Dengue vaccine efficacy” OR “Dengue vaccine safety” OR “Dengue serotypes” OR “Clinical trials” OR “Epidemiological studies”	24
Medline	“Dengue fever” OR “Dengue epidemics” AND “Dengue vaccine” OR “Dengue vaccine development” OR “Dengue vaccine prospects” AND “Dengue vaccine efficacy” OR “Dengue vaccine safety” OR “Dengue serotypes” OR “Clinical trials” OR “Epidemiological studies	210
Health Source	“Dengue fever” OR “Dengue epidemics” AND “Dengue vaccine” OR “Dengue vaccine development” OR “Dengue vaccine prospects” AND “Dengue vaccine efficacy” OR “Dengue vaccine safety” OR “Dengue serotypes”	2
Science Direct	“Dengue fever” OR “Dengue epidemics” AND “Dengue vaccine” OR “Dengue vaccine development” AND “Dengue vaccine efficacy” OR “Dengue vaccine safety” OR “Dengue serotypes” OR “Clinical trials” OR “Epidemiological studies”	1381
Academic Search Premiere	“Dengue fever” OR “Dengue epidemics” AND “Dengue vaccine” OR “Dengue vaccine development” OR “Dengue vaccine prospects” AND “Dengue vaccine efficacy” OR “Dengue vaccine safety” OR “Dengue serotypes” OR “Clinical trials” OR “Epidemiological studies”	134

**Table 2 diseases-12-00032-t002:** Egger’s regression intercept.

Intercept	−0.94387
Standard error	1.22659
95% CI lower limit (2-tailed)	−3.47009
95% CI upper limit (2-tailed)	1.58235
*t*-value	0.76950
df	25
*p*-value (1-tailed)	0.22440
*p*-value (2-tailed)	0.44881

## Data Availability

Available upon request.

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
