# Peer review of "Dengue Fever Epidemics and the Prospect of Vaccines: A Systematic Review and Meta-Analysis Using Clinical Trials in Children"

_diseases, 2024, doi:10.3390/diseases12020032_

Round 1
Reviewer 1 Report
Comments and Suggestions for Authors
The study is important but
1. Throughout the text you mention that the vaccines were effective but how? Please provide evidence. Evidence of development of Ig, types of Ig. Mention the types of immunity: CMI or HIR or both?
2. What is about cytokine profiles> What types of immunity: Th1 or Th2 or mixed.
3. You mentioned that the vaccine was safe but what is the evidence(s). Has the IgE level investigated?
Result is redundant: e.g., sub-heading: 3.4.1 VS 3.3.1 same, among others.
Table 2. is too long. Present as supplemental data.
Discussion: looks like results again. Delete all sub-head and describe the important observation without massive repeatition of results.
Comments on the Quality of English LanguageModerate editing
Reviewer 2 Report
Comments and Suggestions for Authors
‘multiple databases including PubMed, CINAHL, Medline, Health Source, Science Direct, and Academic Search Premier’ – some are platforms hosting content of variable content and impact. Why not opting for WoS and Scopus? In the abstract, font size is variable. Some sources provide close to zero info. E.g., ‘Developing a vaccine that provides broad and durable protection against all four dengue serotypes remains a crucial goal [7]’. In the introduction section, the motivation and objective of the paper shall be further elaborated/discussed. A large proportion of the manuscript needs to be substantially rewritten to provide a more robust analysis of the issues that it raises. The manuscript will benefit from further discussion of pivotal concepts and methodological criteria in order to offer a better articulation between theory and data. The main contributions of the paper should be presented as part of the empirical discussions or critical assessment on the core research outcomes. Some of the cited sources are too old to reflect individually or cumulatively the current picture of a specific topic. You should compare your results with others in terms of concrete data for better research integrative value. What screening and quality assessment tools did you use (e.g., SRDR, Distiller SR, AMSTAR, MMAT, Dedoose, AXIS, ROBIS, etc.) in article selection? To ensure compliance with PRISMA guidelines, a citation software should be used, and at each stage the inclusion or exclusion of articles should be tracked by use of custom spreadsheet. Data visualization tools: VOSviewer (layout algorithms) would optimize your research by harnessing Dimensions (bibliometric mapping) in terms of co-authorship, citation, bibliographic coupling, and co-citation. ‘the studies four studies’ - ? The figures should be improved, unified as style, and thoroughly explained. The conclusion, too short, should clarify the main contribution of the paper and the value added to the field. Conclusion needs to be rewritten so that only important results are brought out along with their interpretation, comparison with earlier studies, and implications in a more integrated fashion.
Comments on the Quality of English Language‘the studies four studies’ - ?
Reviewer 3 Report
Comments and Suggestions for Authors
Please refer to the attached pdf for my comments.

Author Response
Dear Reviewer,
Thank you so much for your kind review and recommendation for the publication of the manuscript. We reviewed your attached comments for each section of the manuscript. They were all positive comments.
Thank you again.
Reviewer 4 Report
Comments and Suggestions for Authors
The present study represents a systematic review and meta-analysis regarding the efficacy and safety of Dengue vaccines. Several methodological concerns were raised, limiting the interpretability of the outcomes.
-The authors should explain the novelty over the existing meta-analyses in the field (e.g.,10.4103/ijcm.IJCM_608_20).
-The protocol of the study was not prospectively registered, e.g., in PROSPERO. Therefore, the meta-analysis was not PRISMA-compliant.
-The search strategy did not include MeSH terms.
-For quality assessment, the ROB-2 tool has been used. The description of the tool in the methods section and the respective citation should be corrected.
- Table 2 is difficult to follow. The numerical outcomes of the included studies should be presented.
-Inter-study heterogeneity was high. The 95% prediction intervals should be calculated in order to present estimates of the effects to be expected by future studies (https://doi.org/10.1136/bmjopen-2015-010247).
-No subgroup analyses were performed. The potential effects of vaccine type, region, study design etc. should be futher explored.
-Publication bias was not evaluated.
-The credibility of evidence should be assessed following the GRADE approach.
Comments on the Quality of English LanguageModerate editing of English language required
Round 2
Reviewer 1 Report
Comments and Suggestions for Authors
Still there are some problem in the presentation style. Please make attractive the subheadings of the results. They look like methods.
Delete all subheadings from discussion
Comments on the Quality of English LanguageMinor checking
Author Response
We addressed all the comments. Our responses are attached.

Reviewer 2 Report
Comments and Suggestions for Authors
This revised version can be published.
Author Response
The Reviewer's comments:
This revised version can be published.
No revisions are suggested.
Thank you so much!
Reviewer 4 Report
Comments and Suggestions for Authors
The raised points were not adequately addressed and some parts of the manuscript, especially the GRADE evaluation are misleading. The 95% prediction intervals were not calculated, subgroup analysis was not performed and publication bias was not statistically assessed. Therefore, the paper cannot be accepted for publication.
Comments on the Quality of English LanguageModerate editing of English language required
Author Response
Dear Reviewer,
Thank you so much for your very useful suggestions, which were addressed point-by-point. We have revised the manuscript accordingly.

Round 3
Reviewer 4 Report
Comments and Suggestions for Authors
The raised points were not adequately addressed (e.g., improper application of GRADE, no true estimation of 95% prediction intervals etc.). Therefore, the manuscript cannot be accepted for publication.
Comments on the Quality of English LanguageModerate editing of English language required